# Frasier Syndrome: A 15-Year-Old Phenotypically Female Adolescent Presenting with Delayed Puberty and Nephropathy

**DOI:** 10.3390/children10030577

**Published:** 2023-03-17

**Authors:** Qing Shao, Xinglei Xie, Jia Geng, Xiaoling Yang, Wei Li, Yuwei Zhang

**Affiliations:** 1Department of Endocrinology and Metabolism, West China Hospital of Sichuan University, Chengdu 610041, China; 2Institute of Rare Diseases, West China Hospital of Sichuan University, Chengdu 610041, China; 3Outpatient Department, West China Hospital of Sichuan University, Chengdu 610041, China

**Keywords:** nephrotic syndrome, focal segmental glomerulosclerosis, delayed puberty, pediatrics, Frasier syndrome, Wilms tumor 1, case report

## Abstract

Frasier syndrome (FS) is a rare inherited disorder characterized by gonadal dysgenesis and progressive nephropathy, resulting from mutations in the intron 9 splice donor site of the Wilms tumor 1 (WT1) gene. It is associated with male gonadal dysgenesis (female external genitalia with a 46 XY karyotype), and a high risk of gonadoblastoma during adolescence. Patients with FS present early in childhood with proteinuria that progressively worsens with a high likelihood of end-stage renal disease (ESRD). Herein, we report a 15-year-old female (karyotype 46, XY) patient characterized by delayed puberty and steroid-resistant nephrotic syndrome, in whom whole genome sequencing showed a mutation in intron 9 of the WT1 gene, c.1447 + 4 C>T. This is the first case of FS with delayed puberty as the first complaint with no previous renal symptoms. We consider delayed puberty as an important manifestation of FS and summarize the diagnostic process of delayed puberty in the female phenotype. For clinicians, delayed puberty is a common disorder in pediatrics but requires vigilance for some rare causes. Etiological screening and chromosome karyotype analysis are important for the early diagnosis of FS in patients with delayed puberty.

## 1. Introduction

Frasier syndrome (FS) is a very rare genetic disorder involving early gonadal and renal development, resulting from a mutation of the Wilms’ tumor 1 (WT1) gene [1]. There are several different syndromes corresponding to mutations in the WT1 gene, including WAGR syndrome with 11p13 deletion [2], Denys-Drash syndrome(DDS) with exons 8 or 9 point mutations [3], and FS with an intron 9 mutation. Frasier syndrome was first reported by Frasier in 1964 [4]. To date, less than 150 cases have been described globally [5]. In China, such reports are more limited [6,7,8].

Frasier syndrome is associated with specific pathogenic variants that affect nucleotides 4–5 of intron 9 in the WT1 gene [2,3]. Nephropathy is the predominant abnormality of this disease. It develops during childhood and presents with persistent proteinuria followed by steroid-resistant nephrotic syndrome (SRNS) [9]. The main characteristics of FS include gonadal dysplasia (female external genitalia with XY sex chromosomes), early adult renal failure, and an increased risk of gonadal tumor. Currently, there is no standard treatment for Frasier syndrome; most treatment is symptomatic and supportive.

This is the first case of FS with delayed puberty as the first complaint with no previous renal symptoms. We retrospectively analyzed the clinical data of the patient and summarized the relevant literature to help clinicians deepen their understanding of FS. In addition, we summarized the diagnostic process of delayed puberty in the female phenotype and emphasized that for patients with delayed puberty, etiologic screening and chromosome karyotype analysis should be used to detect possible genetic abnormalities, thus contributing to the early diagnosis of important genetic conditions, including FS.

## 2. Case Presentation

A 15-year-old phenotypically female adolescent was evaluated by our department of endocrinology and metabolism with delayed puberty and proteinuria. There was no family history of kidney disease or delayed puberty. The patient’s family recounted that she has shown slow growth beginning 3 years previously, with less than 2 cm of net growth per year, accompanied by delayed breast development and lack of pubic or axillary hair. Her intellectual development was deemed normal, and she experienced no other discomfort herself. Therefore, the patient and her family did not attach importance to the symptoms and sought no medical attention. At age 14, the patient presented to the local pediatric endocrinology department for no menstruation and growth retardation. Laboratory tests indicated that the growth hormone (GH) level was 0.31 ng/mL(Normal range(NR) 0.02–5.42), insulin-like growth factor-1 (IGF-1) was 56.8 ng/mL(NR 220.00–972.00), urine albumin (+++), estimated glomerular filtration rate was 56.77 mL/min/1.73 m^2^, and creatinine was 184 umol/L. Sex hormones tests and results are shown in Table 1 (Results 1). The determination of bone age: bone age was about 10.3 years old (bone age lag, less than 3rd percentile). A growth hormone stimulation test was performed with arginine and insulin respectively, and the results showed that the peak GH was 7.77 ng/mL and 7.01 ng/mL. A chromosomal karyotype analysis was not performed. The local hospital considered the patient to have a partial deficiency of GH and recommended growth hormone therapy, which the patient refused. In the same year, she was admitted to the local hospital’s nephrology department with systemic edema, lower back pain, and proteinuria. Laboratory studies indicated nephrotic syndrome: hypoalbuminemia (23.1 g/L), hyperlipidemia (total cholesterol, 9.74 mmol/L), and massive proteinuria (urine albumin (+++)) without hematuria, was detected. She received a renal biopsy, and the pathology (Figure 1) showed focal segmental glomerulosclerosis (FSGS). 

Since the heavy proteinuria did not resolve after 1 year of steroid therapy, the child was transferred to our hospital for further assessment. Physical examination indicated: height: 136 cm (age-specific height < -3SD), BMI: 17.84 kg/m^2^. Waist circumference: 64 cm. Special facial appearance: deformity of the auricles (prominent ears) and a saddle nose. Both breasts were undeveloped, and the areola pigmentation was not obvious; the patient was without armpit and pubic hair. Vulva is of the female type and the vaginal opening is visible, however without clitoris enlargement and posterior lip fusion (Tanner stage 1). The estimated glomerular filtration rate was 22.64 mL/min/1.73 m^2^, creatinine was 262 umol/L, blood albumin 29.0 g/L, urine albumin (++++), and the urine protein quantification was 5.40 g/24 h. The anti–glomerular basement membrane, antinuclear antibody, anti-neutrophil cytoplasmic antibody, complement C3, C4, antistreptolysin-O test and immunoglobulin tests were negative. The parathyroid hormone (PTH) level was 81.98 pmol/L(NR 1.6–6.9 and 25-hydroxy-vitamin D was 15.8 nmol/L(NR 47.7–144). Sex hormone tests and results are shown in Table 1 (Results 2). Chromosome karyotype G-band analysis showed: 46, XY (male karyotype). A doppler ultrasound of the urinary and reproductive system showed no definite ovaria-like or testicular echo detected in the pelvic cavity. A strip of hypoecho was found behind the bladder, with a maximum thickness of about 0.4cm, and no obvious blood flow signal was observed. No tumor or structural abnormality was found on the pituitary MRI. Whole genome sequencing (WGS) of DNA extracted from her peripheral blood was recommended. A reported variation (PMID: 9398852; 9475094; 12050205; 24856380) in the WT1 gene, NM_024426.6:c.1447+4C>T, was detected in this patient. The variation was located in intron 9, which was predicted to destroy the natural splice donor site and led to abnormal gene splicing (PMID:9398852). Her parents and sister were negative for the variation detected with Sanger sequencing (Figure 2). A diagnosis of FS was established, and a bilateral gonadectomy biopsy was performed after informed consent was signed. The uterus was seen to be streaked with dysplasia, and strips of grayish-white dysplastic gonadal tissue were detected in the abdominal cavity bilaterally at the location of the inguinal proximity of the internal ring. The pathological findings were fibrovascular and dysplastic fallopian tubes, and focal gonadal interstitial and follicular structures. No tumorigenic lesions were seen.

The clinical, pathological, and genetic results of the child all supported the diagnosis of FS, and no Wilms tumor or other tumors were found.

We adequately communicated with the patient and family about gender selection and further treatment for the disorders of sexual development. Since the patient was raised as a girl, they decided to remain living as a female, and they refused sex hormone replacement therapy due to their temporary lack of reproductive and sexual needs.

We followed up with the patient by telephone every 6 months after discharge. Proteinuria persisted and the patient’s creatinine had progressed to 2554 umol/L with severe anemia two years after discharge. The patient is presently undergoing regular hemodialysis and preparing for kidney transplantation.

## 3. Discussion

Frasier syndrome is a rare disorder that is inherited in an autosomal dominant pattern. After first being described in 1964, Barbaux et al. detected mutations in WT1 in Frasier syndrome patients [10]. Subsequently, Berta et al. described the clinical overlap and differences between FS and DDS: both had nephrotic syndrome due to FSGS and diffuse mesangial sclerosis; urogenital abnormalities, gonadal risk of blastoma and Wilms tumor, the latter more common in DDS. In addition, DDS presents earlier with SNRS and progresses to ESRD before the age of 2–3 years, while Frasier syndrome has a relatively late onset and a slower progression [11]. Both FS and DDS are induced by mutations in the WT1 tumor suppressor gene, situated on the short arm of chromosome 11. This gene is highly expressed in the nephrogenic zone, proximal tubules, and podocytes, and its normal function is essential for glomerular differentiation, genital development, and tumor suppression [12]. In patients with FS, a donor splice site mutation on WT1 intron 9 (intervening sequence (IVS 9)) results in glomerular lesions, resulting in proteinuria and nephrotic syndrome in childhood and a subsequent progression to renal failure [13]. On the other hand, the reduction of the WT1 + KTS heterodimer, due to the disruption of alternative splicing of the WT1 gene, is associated with a reduced expression of the transcription factors SRY and SOX9 in Sertoli cells, which affects testicular development and leads to gonadal dysplasia [14]. When this gene mutates, it acts dominantly. Most cases are caused by spontaneous mutations, causing the disease in patients with no prior family history [11]. But in rare cases, it can also be inherited from a 46, XX mother to a 46, XY daughter [15,16].

In a systematic review, Ezaki et al. [17] found that except for a few cases with consistent external genitalia and sex chromosome presentation (46, XY with male external genitalia or 46, XX female with external genitalia), 82% of the FS patients were found to have female external genitalia with 46, XY karyotype, of which 50% were diagnosed at puberty due to delayed secondary sexual characteristics (primary amenorrhea and delayed puberty or both). However, since patients with FS mostly presented with nephropathy in childhood, the cases reported have a history of nephropathy or initial symptoms associated with renal disease (e.g., edema, proteinuria, hypertension). Our patient also had a 46, XY karyotype with female genitalia, consistent with other studies, whereas delayed puberty was her initial complaint with no previous renal symptoms. This reflects a significant heterogeneity in the clinical phenotype of the WT1 mutation, which may be related to the variable mosaicism (with different percentages of mosaicism in different tissues) of WT1 with somatic mutations. Of note, this patient did not undergo urinary routine and renal function tests before the evaluation of delayed puberty at the local hospital; elevated PTH and decreased 25-hydroxyvitamin D were found after admission, so it cannot be determined whether she had proteinuria at an early stage or was in a state of subclinical nephropathy. The association of growth retardation with renal disease could not be excluded either. Gurgana et al. [18] and Vidhiya et al. [19] reported cases of FS that demonstrated focal segmental glomerulosclerosis (FSGS) on a renal biopsy, concordant with the kidney biopsy pathology of our patient. However, a different study showed a case of FS with membranoproliferative glomerulonephritis on a renal biopsy. We summarize the clinical features of the previous cases with the same mutation locus (IVS9 +4C>T) as our patient (Appendix A). As shown in the table, among the patients with C>T at the +4 position of WT1 intron 9, 26 individuals with 46, XY karyotype, while only 2 individuals reported the 46, XX karyotype. The reason for this significant difference deserves to be investigated. Patients with karyotype 46, XX presented with normal secondary sexual characteristics and reproductive function [17], symptoms of nephropathy are difficult to distinguish from the common FSGS, which may contribute to a proportion of cases being missed. We also propose another potential hypothesis: a target gene associated with WT1 transcription factors may be present on the X chromosome so that the majority of cases occur in 46, XY individuals and a minority of 46, XX individuals develop nephropathy due to X chromosome inactivation. This is speculation but is important for molecularly-targeted interventions in WT1-related diseases. More studies are needed to confirm it.

Based on the previously reported cases of FS, most studies have recommended genetic testing in patients with a clinical picture comprising gonadal dysgenesis, refractory nephrotic syndrome, resistance to steroid and immunosuppressive medication, or a rapid progression of renal failure. But these manifestations usually imply that long-term diagnosis and treatment are required. 

Delayed puberty is a common disorder in pediatrics, which occurs in approximately 2% of girls and is defined as underdeveloped breasts at a chronological age > -2.5 SD of the population’s average age of pubertal onset. Causes include genetic disorders (Turner’s syndrome, among others.), central nervous system disorders (hypothalamic or pituitary tumors), central nervous system radiation, certain chronic diseases (diabetes with poor glycemic control, inflammatory bowel disease, kidney disease, among others), Kallman syndrome, malnutrition or eating disorders, and excessive physical activity [20]. Delayed puberty is mainly divided into constitutional delay of growth and puberty (CDGP), hypogonadotropic hypogonadism (HH) and hypergonadotropic hypogonadism [21]. CDGP is a non-pathological disorder that is considered to be the most common cause of delayed puberty in adolescents. Patients with CDGP are generally inside the upper limit of the normal variation of pubertal development patterns. Hypogonadotropic hypogonadism is defined by low serum gonadotrophins (LH and FSH) and may be induced by a potential deficiency of the central nervous system, possibly associated with neuroendocrine defects or isolated or idiopathic. Hypergonadotropic hypogonadism is mainly characterized by increased levels of LH and FSH [22]. Howard et al. [23] suggest that a chromosome karyotype analysis should be performed in all hypergonadotropic hypogonadism patients: it can reveal an X0, XX, or XY pattern, or related mosaicism or other variants. Chromosome karyotype XX usually indicates premature ovarian failure, X0 indicates Turner syndrome, and XY indicates sexual developmental disorders (DSD). We have summarized the diagnostic process of delayed puberty in female phenotypes as shown in Figure 3 [21,24,25]. For patients with delayed puberty, it requires vigilance for some rare causes. Regardless of the presence of renal disease, early etiologic screening and chromosome karyotype analysis are necessary to detect possible genetic abnormalities, thus contributing to the early diagnosis of important genetic conditions, including FS.

According to the current few cases, the treatment of FS mainly focuses on the treatment of nephrotic syndrome and gonadal tumors, with no standard treatment for FS in either the nephrology or oncology literature. For nephrotic syndrome, hormones and immunosuppressive [26,27] therapy have been ineffective, thus the clinical manifestation of “refractory nephrotic syndrome”. Administration of the combination of renin-angiotensin system (RAS) inhibitors with indomethacin produced a beneficial anti-proteinuric response in a 2013 case report [28]. Another report concluded that long-term treatment with once-daily low-dose cyclosporine A and RAS inhibition was considered effective in inducing and maintaining partial remission of FS [29]. However, these therapies are controversial due to the paucity of clinical evidence and drug-related side effects. More research is needed to determine effective treatments. Most patients with FS reach end-stage renal disease during adolescence. For these patients, kidney transplantation has been reported to be effective. Compared with other diseases with gonadal dysplasia, patients with FS have a particularly high risk of gonadal tumors (60%) [30]. Gonadoblastoma usually occurs in the second decade of life but can develop as early as 9 months of age in children with gonadal dysgenesis [31]. Hashimoto et al. [13] recommended that patients diagnosed with Frasier syndrome should undergo an early gonadectomy, as performed in our patient. The standard chemotherapy for end-stage malignant germ cell tumors is a combinative treatment of bleomycin, etoposide, and cisplatin. The limitation of cisplatin’s use in renal insufficiency patients is also the reason for recommending an early gonadectomy [13]. For the treatment of disorders of sexual development (DSD), adequate communication and counselling with the patient and family is necessary. Patients can then be helped and supported to make a gender selection and decide whether to undergo surgical treatment. In addition, hormone replacement therapy should be performed to stimulate normal puberty and to promote the development of secondary sexual characteristics. For children with abnormally short height, growth hormone, vitamin D, and calcium, other treatments can be added. In terms of fertility, ovum donation and in vitro fertilization can be considered, and psychological support treatment should not be ignored.

## 4. Conclusions

Frasier syndrome is a rare disease. For a definitive diagnosis of Frasier syndrome, molecular detection should be performed to identify WT1 mutations, which is critical for defining the extent of the disease and proceeding with optimal clinical care. Delayed puberty is a very common disorder in pediatrics, but it requires vigilance for some rare causes such as genetic disorders. For patients presenting with delayed puberty, aggressive etiologic screening and chromosome karyotype analysis may allow clinicians to identify FS earlier by proceeding with genetic testing. However, treatments for FS are limited and more clinical studies are needed.

## Figures and Tables

**Figure 1 children-10-00577-f001:**
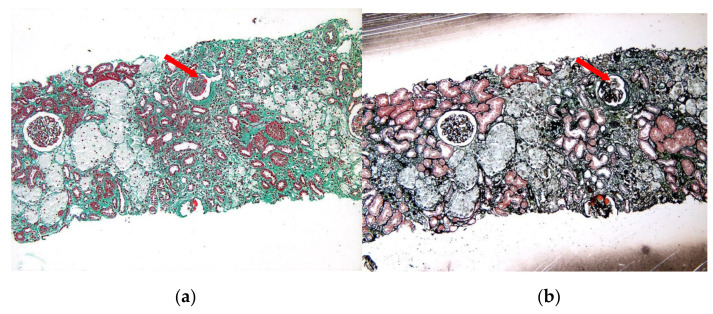
The light microscopy images of the renal biopsy. (**a**) Masson staining; (**b**) PASM staining. The red arrow indicates the focal segmental sclerosis of the glomerulus. Pathological diagnosis: focal and segmental glomerular sclerosis (FSGS) with subacute tubulointerstitial nephritis.

**Figure 2 children-10-00577-f002:**
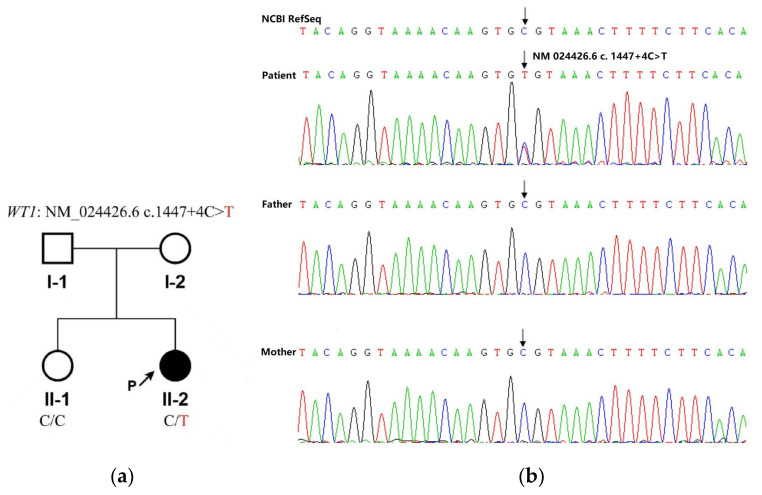
Pedigree (**a**) and sequencing (**b**) analyses. Molecular detection was performed using whole genome sequencing (DNBSEQ-T7) and Sanger sequencing techniques in the affected individual and her parents. The affected child (II-2, indicated by arrow) presented with a single nucleotide substitution in the canonic donor KTS splice site of intron 9 in WT1(IVS9; c.1447 + 4C>T), which was negative in her parents.

**Figure 3 children-10-00577-f003:**
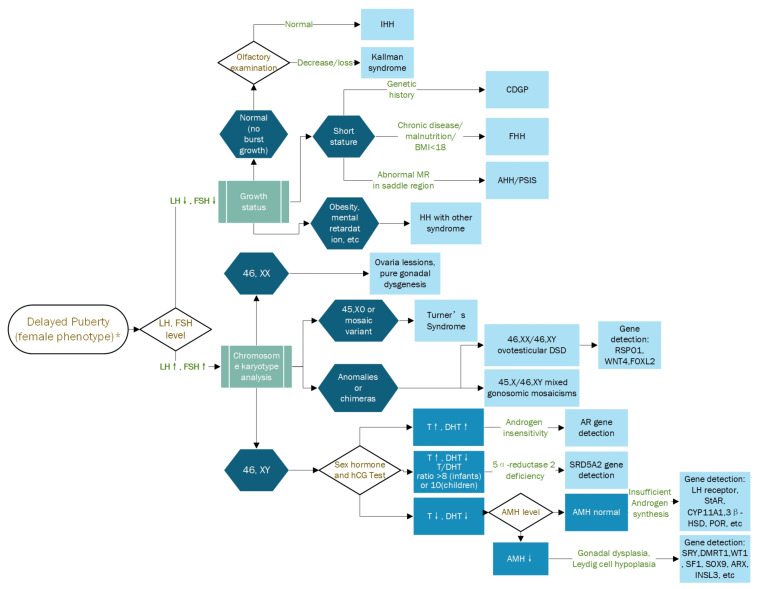
The diagnostic process of delayed puberty in the female phenotype. * Delayed puberty should be considered when: (1) no breast development is observed until age 13; (2) no menarche at age 14, but some degree of breast development; (3) no menarche at age 15 [21]. ↑: higher than normal values. ↓: lower than normal values. LH: luteinizing hormone; FSH: follicle-stimulating hormone; IHH: idiopathic/isolated hypogonadotropic hypogonadism; CDGP: constitutional delay of growth and puberty; FHH: constitutional delay of growth and puberty; AHH: acquired hypogonadotropic hypogonadism; PSIS: pituitary stalk interruption syndrome; HH: hypogonadotropic; Hypogonadism; DSD: disorders of sex development; RSPO1: R-spondin1; WNT4: wingless-type MMTV integration site family, member 4; FOXL2: forkhead box protein L2; hCG: human chorionic gonadotropin; T: testosterone; DHT: dihydrotestosterone; AMH: anti-Müllerian hormone; AR: androgen receptor; SRD5A2: Recombinant Steroid 5 Alpha Reductase 2; StAR: steroidogenic acute regulatory protein; CYP11A1: Cytochrome P450 Family 11 Subfamily A Member 1; CYP17A1: Cytochrome P450 Family 17 Subfamily A Member 1; 3β-HSD: 3β-hydroxysteroid dehydrogenase; POR: Cytochrome P450 Oxidoreductase; SRY: sex-determining region, Y chromosome; DMRT1: double sex- and MAB3-related transcription factor 1; WT1: Wilms’ tumor 1; SF1: steroidogenic factor 1; SOX9: SRY-box transcription factor 9; ARX: Aristaless Related Homeobox; INSL3: insulin-like factor 3.

**Table 1 children-10-00577-t001:** The completed sex hormone tests and results.

Sex Hormones	Results 1	Results 2	Normal Range
PRL (ng/mL)	/	39.60	6.0–29.9
DHEAS (umol/L)	/	1.310	1.77–9.99
LH (IU/L)	76.19	171.0	7.7–58.8
FSH (IU/L)	153.9	>200.0	25.8–134.8
E2 (pg/mL)	5.3	<5.0	<138
P (ng/mL)	0.16	0.23	<0.126
T (ng/mL)	0.06	0.025	0.03–0.27
DHT (ng/mL)	/	0.350	/
AMH (ng/mL)	/	1.05	2–6.8

PRL: Prolactin; DHEAS: Dehydroepiandrosterone sulphate; LH: Luteinizing hormone; FSH: Follicle Stimulating hormone; E2: Estradiol; P: progesterone; T: Testosterone; DHT: Dihydrotestosterone; AMH: anti-Müllerian hormone.

## Data Availability

The data that support the findings of this study are available from the corresponding author upon reasonable request.

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
