# Peer review of "Frasier Syndrome: A 15-Year-Old Phenotypically Female Adolescent Presenting with Delayed Puberty and Nephropathy"

_children, 2023, doi:10.3390/children10030577_

Round 1

Reviewer 1 Report

Summary  

Frasier syndrome: A 15-year-old phenotypically female adolescent presenting with delayed puberty and nephropathy 

This is a case report of a 15 years phenotypically female adolescent with Frasier syndrome presenting with delayed puberty- no menarche at 14 years, growth retardation, and renal failure. The uniqueness of the case is based on the fact that delayed puberty preceded renal failure in this case the reverse is the case in most of the presentations of Frasier syndrome. 

Strength  

A well-written case report with a detailed presentation of the case with images to back up the presentation. 

Clinical management was also well discussed  

Weakness  

The whole case report needs to be abridged – make it shorter and more concise. The reader might lose interest if the report is too long  

Comment  

Please make the discussion shorter.  

The emphasis should be a brief discussion on Frasier syndrome with more emphasis on your case and its uniqueness  

Use arrows to show the areas with focal segmental glomerulosclerosis in Fig 1

Author Response

Thank you for your comments on our manuscript.

Response to Reviewer 1 Comments

Point 1: The whole case report needs to be abridged – make it shorter and more concise. The reader might lose interest if the report is too long 

Comment 

Please make the discussion shorter. 

The emphasis should be a brief discussion on Frasier syndrome with more emphasis on your case and its uniqueness 

Response 1: Thank you for your suggestion, but the manuscript requires a word count of 2500. So, we are very sorry that we may not be able to abridge the content.

Point 2: Use arrows to show the areas with focal segmental glomerulosclerosis in Fig 1

Response 2:  Thank you for your suggestion, and we have added arrows to the Figure 1.

Reviewer 2 Report

Frasier syndrome – case report:    (Paper submitted to Children 2228125)

This case report describes a 15 year-old girl, who after referral and initial investigation for delayed puberty and short stature, presented a few months later with nephrotic syndrome associated with FSGS, leading to genetic investigation and a diagnosis of Frasier syndrome (with XY karyotype).  The authors present this as the first case known to present with symptoms of hypogonadism prior to renal symptoms, and argue that chromosome analysis is an important part of investigation of delayed puberty, in order then to lead on to specific genetic diagnoses, such as Frasier syndrome.

This is a valid conclusion, although perhaps primarily to enable identification of the more common diagnosis of Turner syndrome.  However, because it is surprising that some specialist paediatric endocrinologists are not doing this, the paper here is important for them, in emphasising that this should be done.

There are three major points, and several minor ones requiring attention.

Major Point 1.  The authors need to consider and discuss whether the girl could have had subclinical renal disease for a significant period before referral for delayed puberty. 

From their description, the girl was known to have growth delay for the previous 3 years.  The authors need to indicate whether any investigation was done over that period, and specifically when was the first time that the girl had a routine urinalysis (dipstick) test, or a routine renal function blood test, and whether those showed proteinuria or any other indication of renal disease.  If there is no earlier information, the authors should consider whether there could have been subclinical renal disease, and if so, could this have been contributing to the growth delay, or was this merely the lack of pubertal growth compared with the cumulative increase in the proportion of other girls of the same age going through puberty. 

In case each specific Frasier syndrome mutation may follow slightly different patterns, the authors should also make a Table to compare the clinical data of their case with that of all previously reported cases (of FS and of DDS) with the exact same mutation.  eg. including from Kikuchi et al J.Med Genet. 1998; 35, 45-48  and from Chan et al.  Hong Kong Medical Journal 2006; 12, 225-227.

Major Point 2.

In any girl with delayed puberty and growth delay, it would seem essential to do chromosome analysis at the initial stage of investigation – not so much in expectation of rare genetic syndromes, but for the much commoner Turner syndrome (TS), and its mosaic variants.  TS has an incidence of 1/2000 -1/2500 female births.  If pubertal delay is defined (as here) as lack of breast development at an age >2.5 SD above the mean for this pubertal sign, this represents ~1% of the population. Therefore around 1/20-25 of such girls might be expected to have TS.  It is surprising therefore that chromosome analysis was not done at the initial referral at age 14 years.  The authors need to put greater emphasis on this, and particularly in Figure 3, should consider adding a 4th box on the 4th line, so that ’45,X0 or mosaic variant’ is a separate box alongside the boxes for 46,XY ; 46,XX ; and ‘other chromosome anomalies’.  I note also that the main heading for this Flow chart is ‘Delayed Puberty (Female phenotype)’ so wonder why 47,XXY (Klinefelter syndrome) is currently mentioned with equal prominence as 45,XO, as it would probably be only 47,XXY/45,X0 mosaics that might have a ‘female phenotype’.

Major Point 3.

In discussing the aetiology of Frasier syndrome (line 45: ‘to help clinicians deepen their understanding of FS’), the authors must consider and discuss also the presentation of the same mutation in girls who are 46,XX. They should indicate whether there are any families where a 46,XX mother is heterozygous for the mutation (perhaps with FSGS), and has had 46,XX daughters with FS (thereby accounting for autosomal dominant inheritance), or whether a 46,XX female also has  reproductive problems. They should also comment on the 18% of 46,XY cases who do not have ‘female phenotype’, and whether they have intermediate genital ambiguity.  In practice it would be valuable if the authors could construct and add a simple table to show the expected percentage frequency of the different presentations amongst 46,XX girls, and amongst 46,XY individuals who are heterozygous for a WT1 gene intron 9 donor splice site mutation.

More minor points   

4. Line 37.  The authors need to check whether references 2 & 3 are the correct ones to cite here.  Reference 2 makes no mention of Frasier syndrome, whereas reference 3 does so by citing Barbaux et al 1997 (reference 10 of the present paper).

5. Line 43.  ‘First case’.  See Major point 1 above. In order to be so definite, the authors need to be sure that they have checked all previous clinical reports of Frasier syndrome (particularly those with the exact same mutation), and need to discuss whether this could be  merely that renal symptoms, or at least indications of renal disease, were overlooked, or whether the renal disease really did develop after the pubertal problem started to become evident.  

6. Line 46-49.  I would suggest that the authors take a stronger approach here (see Major Point 2 above).  Rather than ‘…propose …….can be used….thus contributing…FS’  it may be more appropriate to say ‘…emphasise’ (or ‘recommend’)…..’should be used…’ ‘…thus contributing to the early diagnosis of important genetic conditions, including FS.

7. Line 52-3.  This sentence needs a verb.  Eg. ‘There was no….’   Also I would suggest writing:  ‘…kidney disease or delayed puberty…’  (rather than ‘…and…’).

8. Line 57.  At age 14, did the local paediatric endocrinology dept. do a routine urinalysis dipstick test (or any other renal function test).  If so, please indicate the result.  Had the patient’s family doctor ever done one for her on any previous occasion ?

9. Line 77 Figure 1. ‘…the biopsy…’  The Figure Legend should indicate that this is a renal biopsy, and also should mention the features for the reader to note on the biopsy that establish the diagnosis as FSGS.

10. Line 96  This sentence needs a verb.  Eg. ‘…was found…’

11. Line 101-2  ‘…parents and sister were absent with the variation detected…’  This would be better written: ‘…parents and sister were negative for the variation detected…’

12. Line 117-119 and lines 217-218. As a complete case description, the authors could consider including 1-2 sentences here (Immediately before ‘We followed up the patient by telephone….’) to report and discuss the girl’s own reaction to the diagnosis (as a DSD in particular), and her subsequent coping strategy.  This would then link in with the sentence at 217-218 regarding ‘communication’, which at present seems rather bare.  Perhaps this sentence could be:  ‘…adequate communication and counselling…….is necessary.’; followed by:  ’Patients can then be helped and supported to make gender selection….’  

13. Line 124 (and also line 140-1)  ‘There is currently very limited information on this aspect globally’.  What do the authors mean by this ?  Cases in 46,XY females will be due to either de-novo heterozygous mutation, or (presumably) inherited from their mothers (who have 46, XX karyotype).  See Major point 3 above.  

14. Line 126-7.  The authors need to clarify the difference, (as well as the overlap)  between FS and DDS, particularly if the only difference is that Wilms tumour is more common in DDS. This discussion should also be in the context of the presentation in 46,XX girls, as well as in 46,XY individuals (see also points 3 & 13).  

15. Line 143-4  If 82% of FS have female external genitalia with 46,XY karyotype (ref 15), the authors must mention what is the presentation in the other 18% - do these cases have genital ambiguity, and what is their karyotype ?. It would also be helpful if the authors could find data from which to estimate the proportion of 46,XX and separately of 46,XY intron 9 donor-splice-site-mutation-heterozygous individuals who have each of FS, DDS, or other presentation (see points 3,13,14).  

16. Line 144-5.  If 50% of FS are diagnosed at the expected time of puberty due to delayed secondary sexual characteristics, the authors need to say what proportion of these presented with prior symptoms of renal involvement (see Major point 1).

17. Line 158.  ‘Delayed puberty….’    Please start a new paragraph here.

18. Lines 159-160. Please see Major point 2 concerning Turner’s syndrome. This could be mentioned specifically as perhaps the commonest of the ‘genetic disorders’.

19. Line 174.  ‘X0, XX or XY formula etc.’ 
A better term might be ‘X0, XX or XY pattern, or related mosaicism or other variant.’

20. Line 182. Figure 3.  See major point 2.

21. Line 202.  ‘The combination administration….’.   This may be better written as:  ‘Administration of the combination of….’

22. Line 212, and lines 318-320.    ‘Hughes et al [29].’   The currently listed Reference 29 (McCrory et al) is on the subject of head injury in sport, and has no relevance to the present paper.  The authors should ask themselves how it was possible for this reference to appear in this paper, and remedy the fault in the processing. They then need to substitute the correct reference of Hughes et al, which is not listed elsewhere in the references.    

23. Line 217.  ‘disorder’ should be plural.  Ie.  ‘For the treatment of disorders of sexual development….’

24. Line 217-8  ‘…adequate information…’  and  ‘…make gender selection…’   Please see point 12 for suggested rewording.

Author Response

Thank you for your comments on our manuscript.

Response to Reviewer 2 Comments

Point 1: The authors need to consider and discuss whether the girl could have had subclinical renal disease for a significant period before referral for delayed puberty.

From their description, the girl was known to have growth delay for the previous 3 years.  The authors need to indicate whether any investigation was done over that period, and specifically when was the first time that the girl had a routine urinalysis (dipstick) test, or a routine renal function blood test, and whether those showed proteinuria or any other indication of renal disease.  If there is no earlier information, the authors should consider whether there could have been subclinical renal disease, and if so, could this have been contributing to the growth delay, or was this merely the lack of pubertal growth compared with the cumulative increase in the proportion of other girls of the same age going through puberty. In case each specific Frasier syndrome mutation may follow slightly different patterns, the authors should also make a Table to compare the clinical data of their case with that of all previously reported cases (of FS and of DDS) with the exact same mutation.  eg. including from Kikuchi et al J.Med Genet. 1998; 35, 45-48  and from Chan et al.  Hong Kong Medical Journal 2006; 12, 225-227.

Response 1: Thank you for your suggestion. We considered that it is indeed possible that the patient had subclinical nephropathy. She did not undergo urinary routine and renal function tests before her evaluation for delayed puberty at the local hospital, so it was not known whether proteinuria was present early. Hyperparathyroidism secondary to chronic kidney disease should be considered based on indicators such as PTH and 25-hydroxy-vitamin D after admission, so we could not exclude that the growth retardation was correlated with kidney disease. We summarized all cases with IVS9+4C>T mutations of WT1 gene, as shown in Supplemental Table 1.

Point 2: In any girl with delayed puberty and growth delay, it would seem essential to do chromosome analysis at the initial stage of investigation – not so much in expectation of rare genetic syndromes, but for the much commoner Turner syndrome (TS), and its mosaic variants.  TS has an incidence of 1/2000 -1/2500 female births.  If pubertal delay is defined (as here) as lack of breast development at an age >2.5 SD above the mean for this pubertal sign, this represents ~1% of the population. Therefore around 1/20-25 of such girls might be expected to have TS.  It is surprising therefore that chromosome analysis was not done at the initial referral at age 14 years.  The authors need to put greater emphasis on this, and particularly in Figure 3, should consider adding a 4th box on the 4th line, so that ’45,X0 or mosaic variant’ is a separate box alongside the boxes for 46,XY ; 46,XX ; and ‘other chromosome anomalies’.  I note also that the main heading for this Flow chart is ‘Delayed Puberty (Female phenotype)’ so wonder why 47,XXY (Klinefelter syndrome) is currently mentioned with equal prominence as 45,XO, as it would probably be only 47,XXY/45,X0 mosaics that might have a ‘female phenotype’.

Response 2: Thank you for your suggestion. We have modified Figure 3. Including adding a 4th box ‘45,X0 or mosaic variant’ and deleting 47,XXY (Klinefelter syndrome).

Point 3: In discussing the aetiology of Frasier syndrome (line 45: ‘to help clinicians deepen their understanding of FS’), the authors must consider and discuss also the presentation of the same mutation in girls who are 46,XX. They should indicate whether there are any families where a 46,XX mother is heterozygous for the mutation (perhaps with FSGS), and has had 46,XX daughters with FS (thereby accounting for autosomal dominant inheritance), or whether a 46,XX female also has  reproductive problems. They should also comment on the 18% of 46,XY cases who do not have ‘female phenotype’, and whether they have intermediate genital ambiguity.  In practice it would be valuable if the authors could construct and add a simple table to show the expected percentage frequency of the different presentations amongst 46,XX girls, and amongst 46,XY individuals who are heterozygous for a WT1 gene intron 9 donor splice site mutation.

Response 3: Thank you for your suggestion. We have supplemented the manuscript with a description of the rare genetic situation of FS (46,XX mother inherited to 46,XY daughter) and the phenotypic situation of the remaining 18% of patients in 46,XY cases. Since FS cases with 46,XX karyotype are very rare with different mutation loci and clinical manifestations, and there are more than 70 cases of 46,XY, we did not calculate the expected percentage frequency of different manifestations.

More minor points  

  1. Line 37. The authors need to check whether references 2 & 3 are the correct ones to cite here. Reference 2 makes no mention of Frasier syndrome, whereas reference 3 does so by citing Barbaux et al 1997 (reference 10 of the present paper).

Response 4: Thank you for your suggestion. We have checked references 2 and 3, where reference 2 is about WAGR syndrome and reference 3 is about DDS, both of which are caused by mutations of the WT1 gene. The introduction section cites these two references to describe the diseases associated with WT1 gene and they are not misquoted.

  1. Line 43. ‘First case’. See Major point 1 above. In order to be so definite, the authors need to be sure that they have checked all previous clinical reports of Frasier syndrome (particularly those with the exact same mutation), and need to discuss whether this could be  merely that renal symptoms, or at least indications of renal disease, were overlooked, or whether the renal disease really did develop after the pubertal problem started to become evident. 

Response 5: Thank you for your suggestion. We have checked the previous literature and this is indeed the first case of FS with delayed puberty as the first complaint with no previous renal symptoms. We consider that it is possible for the patient to have subclinical nephropathy without corresponding symptomatic manifestations, so we have added a discussion of this condition.

  1. Line 46-49. I would suggest that the authors take a stronger approach here (see Major Point 2 above). Rather than ‘…propose …….can be used….thus contributing…FS’  it may be more appropriate to say ‘…emphasise’ (or ‘recommend’)…..’should be used…’ ‘…thus contributing to the early diagnosis of important genetic conditions, including FS.

Response 6: Thank you for your suggestion, and we have made the correction.

  1. Line 52-3. This sentence needs a verb. Eg. ‘There was no….’   Also I would suggest writing:  ‘…kidney disease or delayed puberty…’  (rather than ‘…and…’).

Response 7: Thank you for your suggestion, and we have made the correction.

  1. Line 57. At age 14, did the local paediatric endocrinology dept. do a routine urinalysis dipstick test (or any other renal function test). If so, please indicate the result.  Had the patient’s family doctor ever done one for her on any previous occasion ?

Response 8: Thank you for your suggestion. The patient had no previous kidney-related investigations, and it was the first time that urinary protein and renal hypofunction were detected at the local endocrinology department. We have added the corresponding test results in the manuscript.

  1. Line 77 Figure 1. ‘…the biopsy…’ The Figure Legend should indicate that this is a renal biopsy, and also should mention the features for the reader to note on the biopsy that establish the diagnosis as FSGS.

Response 9: Thank you for your suggestion, and we have made the correction.

  1. Line 96 This sentence needs a verb. Eg. ‘…was found…’

Response 10: Thank you for your suggestion, and we have made the correction.

  1. Line 101-2 ‘…parents and sister were absent with the variation detected…’ This would be better written: ‘…parents and sister were negative for the variation detected…’

Response 11: Thank you for your suggestion, and we have made the correction.

  1. Line 117-119 and lines 217-218. As a complete case description, the authors could consider including 1-2 sentences here (Immediately before ‘We followed up the patient by telephone….’) to report and discuss the girl’s own reaction to the diagnosis (as a DSD in particular), and her subsequent coping strategy. This would then link in with the sentence at 217-218 regarding ‘communication’, which at present seems rather bare. Perhaps this sentence could be:  ‘…adequate communication and counselling…….is necessary.’; followed by:  ’Patients can then be helped and supported to make gender selection….’ 

Response 12: Thank you for your suggestion, and we have made the correction.

  1. Line 124 (and also line 140-1) ‘There is currently very limited information on this aspect globally’. What do the authors mean by this ?  Cases in 46,XY females will be due to either de-novo heterozygous mutation, or (presumably) inherited from their mothers (who have 46, XX karyotype).  See Major point 3 above. 

Response 13: Thank you for your suggestion. We have deleted the sentence‘There is currently very limited information on this aspect globally’, and supplemented the manuscript with a description of the rare genetic situation of FS (46,XX mother inherited to 46,XY daughter).

  1. Line 126-7. The authors need to clarify the difference, (as well as the overlap) between FS and DDS, particularly if the only difference is that Wilms tumour is more common in DDS. This discussion should also be in the context of the presentation in 46,XX girls, as well as in 46,XY individuals (see also points 3 & 13). 

Response 14: Thank you for your suggestion, and we have made the correction.

  1. Line 143-4 If 82% of FS have female external genitalia with 46,XY karyotype (ref 15), the authors must mention what is the presentation in the other 18% - do these cases have genital ambiguity, and what is their karyotype ?. It would also be helpful if the authors could find data from which to estimate the proportion of 46,XX and separately of 46,XY intron 9 donor-splice-site-mutation-heterozygous individuals who have each of FS, DDS, or other presentation (see points 3,13,14).

Response 15: Thank you for your suggestion. We have supplemented the manuscript with the phenotypic situation of the remaining 18% of patients in 46,XY cases. Since FS cases with 46,XX karyotype are very rare with different mutation loci and clinical manifestations, and there are more than 70 cases of 46,XY, we did not calculate the expected percentage frequency of different manifestations.

  1. Line 144-5. If 50% of FS are diagnosed at the expected time of puberty due to delayed secondary sexual characteristics, the authors need to say what proportion of these presented with prior symptoms of renal involvement (see Major point 1).

Response 15: Thank you for your suggestion. We have checked the previous literature and all these patients presented with previous symptoms of kidney involvement.

  1. Line 158. ‘Delayed puberty….’ Please start a new paragraph here.

Response 17: Thank you for your suggestion, and we have made the correction.

  1. Lines 159-160. Please see Major point 2 concerning Turner’s syndrome. This could be mentioned specifically as perhaps the commonest of the ‘genetic disorders’.

Response 18: Thank you for your suggestion, and we have made the correction.

  1. Line 174. ‘X0, XX or XY formula etc.’

A better term might be ‘X0, XX or XY pattern, or related mosaicism or other variant.’

Response 19: Thank you for your suggestion, and we have made the correction.

  1. Line 182. Figure 3. See major point 2.

Response 20: Thank you for your suggestion, and we have made the correction.

  1. Line 202. ‘The combination administration….’. This may be better written as:  ‘Administration of the combination of….’

Response 21: Thank you for your suggestion, and we have made the correction.

  1. Line 212, and lines 318-320. ‘Hughes et al [29].’ The currently listed Reference 29 (McCrory et al) is on the subject of head injury in sport, and has no relevance to the present paper.  The authors should ask themselves how it was possible for this reference to appear in this paper, and remedy the fault in the processing. They then need to substitute the correct reference of Hughes et al, which is not listed elsewhere in the references.   

Response 22: We are very sorry for our mistake. We have replaced the correct reference. Thank you for your reminding.

  1. Line 217. ‘disorder’ should be plural. Ie.  ‘For the treatment of disorders of sexual development….’

Response 23: Thank you for your suggestion, and we have made the correction.

  1. Line 217-8 ‘…adequate information…’ and  ‘…make gender selection…’   Please see point 12 for suggested rewording.

Response 24: Thank you for your suggestion, and we have made the correction.

Reviewer 3 Report

Major Comments:

1.   Please mention the importance of the study in the Introduction?

2.   What is the prevalence of FS in the global population? The authors may include information on prevalence of FS in different ethnic group, and correlation of FS with gender and age if any in the ‘Introduction’.

3.   Why nephropathy and delayed puberty are associated with FS?

4.   Delayed puberty is associated with different genetic diseases. Therefore, how do the authors indicate delayed puberty as a prognostic marker of FS?

5.   What are the treatment options available for FS and how does those options can help in extending the survivability of the FS patients detected earlier?

Minor Comments:

1.     The manuscript should be thoroughly checked for the grammatical errors.

Author Response

Thank you for your comments on our manuscript.

Response to Reviewer 3 Comments

Point 1: Please mention the importance of the study in the Introduction?

Response 1: Thank you for your suggestion. We have modified the ‘Introduction’ as appropriate to highlight the importance of this study.

Point 2: What is the prevalence of FS in the global population? The authors may include information on prevalence of FS in different ethnic group, and correlation of FS with gender and age if any in the ‘Introduction’.

Response 2: Thank you for your suggestion. FS is a rare disease, with fewer than 150 cases described worldwide to date, so there are no data on prevalence yet. Also, no literature is available on the prevalence of FS in different ethnic group. For the correlation of FS with gender and age, we described it in ‘Introduction’ as ’It develops during childhood and presents with persistent proteinuria followed by steroid-resistant nephrotic syndrome (SRNS). The main characteristics of FS include gonadal dysplasia (female external genitalia with XY sex chromosomes), early adult renal failure, and an increased risk of gonadal tumor.’

Point 3: 3. Why nephropathy and delayed puberty are associated with FS?

Response 3: Thank you for your question. We described the pathogenesis of FS in ‘Discussion’ as ‘In patients with FS, a point mutation on donor splice site on intron 9 [intervening sequence (IVS 9)] of WT1 leads to nephropathy which appears later in life and is due to focal segmental glomerulosclerosis. On the other hand, a decrease in WT1 + KTS isoforms due to disruption of alternative splicing of the WT1 gene is associated with diminished expression of the transcription factors SRY and SOX9 in Sertoli cells, thereby impairing testicular development and causing gonadal dysplasia’. Delayed puberty is a manifestation of gonadal dysgenesis. Since FS patients have female external genitalia and 46, XY karyotype, they manifest as 'delayed puberty' such as non-development of secondary sexual characteristics and non-menstruation.

  1. Delayed puberty is associated with different genetic diseases. Therefore, how do the authors indicate delayed puberty as a prognostic marker of FS?

Response 4: Thank you for your advice, it is true that delayed puberty cannot be used as an indicator of prognosis for FS. We have revised in the manuscript as ’for patients with delayed puberty, etiologic screening and chromosome karyotype analysis should be used to detect possible genetic abnormalities in patients, thus contributing to the early diagnosis of important genetic conditions, including FS.’

  1. What are the treatment options available for FS and how does those options can help in extending the survivability of the FS patients detected earlier?

Response 5: Thank you for your question. There is no standard treatment for FS yet. For nephropathy, there are reports on the combination of renin-angiotensin system (RAS) inhibitors with indomethacin to reduce urinary protein and long-term treatment with once-daily low-dose cyclosporine A and RAS inhibition is effective in inducing and maintaining partial remission of FS. However, these therapies are controversial due to the paucity of clinical evidence and drug-related side effects. More research is needed to determine effective treatments. For gonadal dysgenesis, the main treatments include gonadectomy to prevent gonadal tumors, sex selection and hormone replacement therapy. We have reviewed the effectiveness and limitations of the above therapies in ‘Discussion’.

Round 2

Reviewer 2 Report

The authors have attended to the points made in the first review, and the paper now reads well.  In particular, the authors do now provide a Supplementary table with details of previous literature-reported cases of FS, DDS or nephropathy who have exactly the same WT1 mutation. However, there is one additional case (of 46,XY Frasier Syndrome) in the publication of Barbaux et al (ref 20 of the Suppl. Table) which has been missed from the Supplementary Table  and this should also be included. 

With the data in the Supplementary Table, it is important that the authors point out that (including this extra case and their own case) there are : 26 reported 46,XY individuals with ‘C>T at position +4 of intron 9 of WT1’, but only 2 reports of ‘C>T at position +4 of intron 9’ in 46,XX individuals.  Part of this is clearly a bias towards ascertainment of cases with genital ambiguity or sex-reversal (and delayed puberty). This in itself is important for the authors to point out, as it implies there would be several missing cases of otherwise normal girls who develop FSGS, where there would be potential genetic implications for any children of theirs. The authors should comment specifically on this.

However, in doing so, they should also consider whether an additional potential explanation, could be a true bias of renal manifestation towards 46,XY individuals if one of the target genes for the WT1 transcription factor were to be an X-linked gene.  This would be important information for establishing the pathogenetic mechanism of the development of FSGS in WT1 mutation.  The 46,XX individuals who do manifest renal disease could potentially then be ones with skewed X-inactivation, which would be a potentially testable hypothesis, and important for molecularly-targeted intervention research. It would also mean that one could expect a wide range of age at onset and severity of renal disease in any 46,XX females who do develop FSGS with the WT1 ‘C>T at position +4 of intron 9’ mutation.

In addition, the authors should also comment, in relation to the relatively late onset of renal symptoms in their own case, that since theirs and almost all other cases of ‘C>T at position +4 of intron 9 of WT1’ (including the two 46,XX cases) have been de novo, whether variable mosaicism from somatic mutation (with different percentages of mosaicism in different tissues) could  potentially account for later-onset cases, even if this is not evident in their peripheral blood sample. 

There are also still a couple of improvements which could be made in the writing :

Line 57:  ‘…development and lack of…’

Line 66:  ‘…3rd percentile…’

Line 69:  ‘…to have partial deficiency…’

Line 211:   I presume that the first version of this diagram is the one to be deleted.

Author Response

Thank you for your comments on our manuscript.

Point 1: The authors have attended to the points made in the first review, and the paper now reads well.  In particular, the authors do now provide a Supplementary table with details of previous literature-reported cases of FS, DDS or nephropathy who have exactly the same WT1 mutation. However, there is one additional case (of 46,XY Frasier Syndrome) in the publication of Barbaux et al (ref 20 of the Suppl. Table) which has been missed from the Supplementary Table 1 and this should also be included.

Response 1: Thank you for your kind reminder. We have added the case to Supplementary Table 1.

Point 2: With the data in the Supplementary Table, it is important that the authors point out that (including this extra case and their own case) there are : 26 reported 46,XY individuals with ‘C>T at position +4 of intron 9 of WT1’, but only 2 reports of ‘C>T at position +4 of intron 9’ in 46,XX individuals.  Part of this is clearly a bias towards ascertainment of cases with genital ambiguity or sex-reversal (and delayed puberty). This in itself is important for the authors to point out, as it implies there would be several missing cases of otherwise normal girls who develop FSGS, where there would be potential genetic implications for any children of theirs. The authors should comment specifically on this.

However, in doing so, they should also consider whether an additional potential explanation, could be a true bias of renal manifestation towards 46,XY individuals if one of the target genes for the WT1 transcription factor were to be an X-linked gene.  This would be important information for establishing the pathogenetic mechanism of the development of FSGS in WT1 mutation.  The 46,XX individuals who do manifest renal disease could potentially then be ones with skewed X-inactivation, which would be a potentially testable hypothesis, and important for molecularly-targeted intervention research. It would also mean that one could expect a wide range of age at onset and severity of renal disease in any 46,XX females who do develop FSGS with the WT1 ‘C>T at position +4 of intron 9’ mutation.

Response 2: Thank you for your suggestion. We agree with your comments and we have analyzed the reasons for bias of renal manifestation for 46, XY individuals in ‘Discussion’: As shown in the table, among the patients with C>T at the +4 position of WT1 intron 9, 26 individuals with 46, XY karyotype, while only 2 individuals reported the 46, XX karyotype. The reason for this significant difference deserves to be investigated. Since patients with karyotype 46, XX presented with normal secondary sexual characteristics and reproductive function, while symptoms of nephropathy are difficult to distinguish from the common FSGS, which may contribute to a proportion of cases being missed. We also propose another potential hypothesis that a target gene associated with WT1 transcription factors may be present on the X chromosome, so that majority of cases occur in 46, XY individuals and a minority of 46, XX individuals develop nephropathy due to X chromosome inactivation. This is a speculation, but is important for molecularly-targeted interventions in WT1-related diseases. More studies are needed to confirm it.

Point 3: In addition, the authors should also comment, in relation to the relatively late onset of renal symptoms in their own case, that since theirs and almost all other cases of ‘C>T at position +4 of intron 9 of WT1’ (including the two 46,XX cases) have been de novo, whether variable mosaicism from somatic mutation (with different percentages of mosaicism in different tissues) could  potentially account for later-onset cases, even if this is not evident in their peripheral blood sample.

Response 3: Thank you for your suggestion. We agree with your comments and add an analysis of the possible causes of the late onset of renal symptoms in our patient, including the heterogeneity of clinical symptoms due to mutations in the WT1 gene and variable mosaicism from somatic mutation.

Point 4: There are also still a couple of improvements which could be made in the writing :

Line 57:  ‘…development and lack of…’

Line 66:  ‘…3rd percentile…’

Line 69:  ‘…to have partial deficiency…’

Line 211:   I presume that the first version of this diagram is the one to be deleted.

Response 4 : Thank you for your suggestion, and we have made the correction.

Reviewer 3 Report

The manuscript is now improved significanly. It needs the thorough checking of the grammar and typos.

Author Response

Thank you for your comments on our manuscript. We have checked the manuscript for grammar and typos and made the correction.